# SPACR Encoded by *IMPG1* Is Essential for Photoreceptor Survival by Interplaying between the Interphotoreceptor Matrix and the Retinal Pigment Epithelium

**DOI:** 10.3390/genes13091508

**Published:** 2022-08-23

**Authors:** Guillaume Olivier, Philippe Brabet, Nelly Pirot, Morgane Broyon, Laurent Guillou, Chantal Cazevieille, Chamroeun Sar, Melanie Quiles, Emmanuelle Sarzi, Marie Pequignot, Ervann Andreo, Agathe Roubertie, Isabelle Meunier, Agnès Muller, Vasiliki Kalatzis, Gaël Manes

**Affiliations:** 1Institute for Neurosciences of Montpellier (INM), Institut National de la Santé et de la Recherche Médicale (INSERM), University of Montpellier, 34091 Montpellier, France; 2RHEM, IRCM-INSERM, 34000 Montpellier, France; 3INSERM, University Claude Bernard, 69100 Lyon, France; 4INM, University of Montpellier, INSERM, CHU Montpellier, 34000 Montpellier, France

**Keywords:** *IMPG1*, *IMPG2*, SPACR, SPACRCAN, vitelliform macular dystrophy, retinitis pigmentosa

## Abstract

Several pathogenic variants have been reported in the *IMPG1* gene associated with the inherited retinal disorders vitelliform macular dystrophy (VMD) and retinitis pigmentosa (RP). *IMPG1* and its paralog *IMPG2* encode for two proteoglycans, SPACR and SPACRCAN, respectively, which are the main components of the interphotoreceptor matrix (IPM), the extracellular matrix surrounding the photoreceptor cells. To determine the role of SPACR in the pathological mechanisms leading to RP and VMD, we generated a knockout mouse model lacking *Impg1*, the mouse ortholog. *Impg1*-deficient mice show abnormal accumulation of autofluorescent deposits visible by fundus imaging and spectral-domain optical coherence tomography (SD-OCT) and attenuated electroretinogram responses from 9 months of age. Furthermore, SD-OCT of *Impg1*^−/−^ mice shows a degeneration of the photoreceptor layer, and transmission electron microscopy shows a disruption of the IPM and the retinal pigment epithelial cells. The decrease in the concentration of the chromophore 11-*cis*-retinal supports this loss of photoreceptors. In conclusion, our results demonstrate the essential role of SPACR in maintaining photoreceptors. *Impg1*^−/−^ mice provide a novel model for mechanistic investigations and the development of therapies for VMD and RP caused by *IMPG1* pathogenic variants.

## 1. Introduction

Within the central nervous system, the extracellular matrix (ECM) is a highly organized structure that surrounds neuronal cells and plays a prominent role in development and homeostasis [1]. The so-called interphotoreceptor matrix (IPM) is the ECM surrounding the cone and rod photoreceptor cells of the retina. The IPM extends throughout the subretinal space from the Müller cells to the apical surface of the underlying retinal pigment epithelium (RPE) [2]. The IPM has numerous functions, including the regulation of retinoid transport, oxygen and nutrient exchange, as well as photoreceptor disk turnover, and it also contributes to retinal adhesion, cell interactions, and photoreceptor alignment and maintenance [2].

The *IMPG1* (Interphotoreceptor Matrix ProteoGlycan 1) gene and its paralog, *IMPG2,* encode SPACR (SialoProtein Associated with Cones and Rods) and SPACRCAN (SialoProtein Associated with Cones and Rods proteoglyCANs), respectively, two major components of the IPM synthesized by the photoreceptors [3,4]. SPACR is a 150-kDa secreted glycoprotein that contains a central mucin-like domain, a carboxy-terminal EGF-like (Epidermal Growth Factor) domain, and two SEA (Sea Urchin Sperm Protein, Enterokinase, and Agrin) domains [5] (Figure 1A). Although the exact function of the SEA domains remains to be elucidated, they may play a role in the cleavage of SPACR at different sites of the protein and thus contribute to the adaptive capacity of the IPM [6]. SPACR is highly homologous to SPACRCAN, a 200-kDa transmembrane photoreceptor proteoglycan [4,7,8]. SPACR and SPACRCAN, two retina-specific proteins, play an important role in the maturation of the photoreceptor outer segments (OS) [9]. Both proteins bind to two of the main glycosaminoglycan molecules essential for IPM organization: chondroitin sulfate and hyaluronan [8,10].

The IPM also plays an important role in the etiology of inherited retinal disorders (IRDs) [2]. Mutations in *IMPG1* and *IMPG2* have been previously associated with two clinically distinct IRDs, vitelliform macular dystrophy (VMD) and retinitis pigmentosa (RP) [11,12,13,14]. VMD is predominantly characterized by central visual loss and an accumulation of autofluorescent material in the macular subretinal space [15]. RP, the most prevalent IRD, is characterized by bone-spicule pigment deposits in the peripheral retina, attenuated retinal blood vessels, night blindness, and progressive visual field loss in a concentric pattern; central vision tends to be preserved until the later stages of disease evolution [16].

Until recently, the role of SPACR and SPACRCAN in the pathophysiology of these two IRDs was unclear due to the lack of disease-specific animal models. In an effort to circumvent this limitation, we recently investigated the role of both proteins in medaka fish following morpholino-mediated ablation of *Impg1* and *Impg2.* We observed a decrease in the length of rod and cone photoreceptor OS, which is consistent with the clinical phenotype of affected RP patients [11]. Similarly, a knockout *Impg2*^−/−^ mouse model was recently generated and shown to mimic the clinical phenotype observed in RP patients [17]. Lastly, an independent team created three murine models, *Impg1*^−/−^*, Impg2*^−/−^, and a double knockout *Impg1*^−/−^*/Impg2*^−/−^ [18]. Surprisingly, no pathological signs were detected in the *Impg1*^−/−^ model or in the double knockout model. By contrast, *Impg2*^−/−^ mice showed retinal lesions that were visible by spectral-domain optical coherence tomography (SD-OCT) and histochemical studies, as well as decreased electroretinogram (ERG) responses at 8 months of age [18].

In this study, we generated and characterized a novel *Impg1*^−/−^ mouse model. *Impg1*^−/−^ mice exhibit progressive accumulation of autofluorescent material forming vitelliform deposits between the RPE and photoreceptor OS, a thinning of the outer nuclear layer, and late signs of photoreceptor degeneration, as determined by attenuated scotopic and photopic ERG responses. Thus, in our animal model, the absence of *Impg1* impairs the IPM, leading to the death of photoreceptors and the underlying RPE, resulting in a progressive loss of visual function.

## 2. Materials and Methods

### 2.1. Generation of the Impg1^−/−^ Mouse Model

The *Impg1*^−/−^ mouse model was generated by gene targeting using homologous recombination. The exon 6 of *Impg1* was floxed in C57Bl/6J embryonic stem cells to disrupt the open reading frame. After successive crossings with a CMV-CRE recombinase mouse model, followed by backcrossing with the C57Bl/6J strain (Jackson Laboratory), we obtained the *Impg1*^−/−^ colony. The mice were maintained under 12 h light (90 lux)/dark cycles with food and water provided *ad libitum* at the Institute for Neurosciences of Montpellier animal facility. The housing conditions were pathogen-free, kept at a constantly controlled humidity (82%), and at a regulated temperature (22 °C). Both sexes were used in all experiments. All experimental protocols involving mice in this study were approved by the French national ministry of higher education and research. All methods were carried out in accordance with relevant guidelines and regulations. All methods are reported in accordance with the ARRIVE guidelines.

### 2.2. Western Blotting Analysis

Whole retinas dissected from wild-type or *Impg1*^−/−^ mice were ground and lysed in RIPA buffer (Thermo Fisher, Illkirch, France) containing protease and phosphatase inhibitor cocktail tablets (Roche Inc., Boulogne-Billancourt, France). An Enhanced BCA Protein Assay kit (Beyotime Biotechnology, Shanghai, China) was used for protein quantification. Proteins (20 μg) were loaded onto a 10% polyacrylamide gel and analyzed by immunoblotting after transfer onto nitrocellulose (NC) membranes. Membranes were saturated within blocking buffer, 1X Tris-buffered Saline 0.1% Tween-20 (TBST) with 5% milk for 2 h, then incubated with primary antibodies (diluted in blocking buffer) overnight at 4 °C. The NC membranes were washed 3 times with 1X TBST buffer for 10 min and incubated with secondary antibodies (diluted in TBST 5% milk) for 1.5 h at room temperature. Antibodies used for Western blotting were anti-SPACR (Santa Cruz, Nanterre, France, Cat N° 377366, 2 µg/mL dilution) and anti-GAPDH (Sigma-Aldrich, Saint-Quentin-Fallavier, France, 0.05 µg/mL dilution). Primary antibodies were detected with goat anti-mouse (Santa Cruz, Nanterre, France, Cat N° sc-2005, 1:5000 dilution) or goat anti-rabbit (Santa Cruz, Nanterre, France, Cat N° sc-2030, 1:5000 dilution) HRP-conjugated secondary antibodies and Clarity Western ECL Substrate (BioRad, Grabels, France, Cat N° 1705060). Western blots were analyzed with the ChemiDoc XRS + Gel Imaging System with Image Lab Software (BioRad, Grabels, France, Cat N° 708265).

### 2.3. In Vivo Imaging

Eyes were dilated with 0.5% tropicamide (Mydriaticum, Théa, France). Mice were anesthetized prior to imaging with a ketamine and xylazine solution (80 mg/kg ketamine, 16 mg/kg xylazine, (Bayer Healthcare, Lyon, France)). An ophthalmic gel (Lacryvisc; Alcon^®^, Novartis; Fort Worth, TX, USA) was applied to keep the cornea moist and minimize the refraction of light. Mice were kept on a heating plate throughout the procedures. Fundus imaging was performed using a Retinal Imaging System (Phoenix Micron III, California, CA, USA). Images were obtained with the Discover software (Phoenix research lab; Phoenix, AZ, USA). Autofluorescence filters were developed with an excitation range of 444.5–564.5 nm and an emission range above 568 nm. SD-OCT was performed on 2-, 9-, and 14-month-old mice using an EnVisu R2200 imaging (Bioptigen, Leica Microsystems; Wetzlar, Germany), as previously described [19]. The images covered a surface area of 1.4 × 1.4 mm^2^ centered on the optic nerve with 100 b-scans per eye with 1000 points resolution per b-scan. For quantitative data, we performed an average of the ratios of outer to inner retinal layer thickness (10 measures per retina; *n* = 5 mice per group).

### 2.4. Electroretinogram (ERG) Recordings

ERG recordings were performed on mice aged 2, 9, and 14 months. Age-matched control and mutant mice were dark-adapted overnight, and all subsequent procedures were performed under dim red light. Following anesthesia, the pupils of the mice were dilated with a drop of 0.5% tropicamide, 0.4% oxybuprocaine hydrochloride, and 10% phenylephrine. Eyes were moistened regularly throughout the procedure to avoid dryness (Lacryvisc; Alcon, Novartis; Fort Worth, TX, USA) and to maintain low impedances. Mice were placed on a heating plate with a rectal temperature control probe. The reference electrodes were placed subcutaneously, behind the ears and at the lower back of the mouse. ERGs were recorded with cotton electrodes soaked in physiological saline, as previously described [20], using Visiosystem equipment (SIEM Bio-Médicale; Nîmes, France). Full scotopic flash intensity ERG was performed in response to light flashes (5 ms) with intensities ranging from −1 to 2 log cd⋅s⋅m^−2^. The measurements corresponded to the average of 7 light flashes with a frequency of 0.3 Hz. For photopic ERG, a white background light of 2 log cd⋅s⋅m^−2^ was used to light adapt mice for 5 min. Then, different light stimulations were used: blue flashes at 2 log cd⋅s⋅m^−2^ intensities and green flashes at 2 log cd⋅s⋅m^−2^, and the last phase corresponded to the flicker protocol: achromatic flashes of 2 log cd⋅s⋅m^−2^ at a frequency of 10 Hz. For flicker recordings, measurements corresponded to the average of the amplitudes of the 5 waves visible on a window of 500 ms. Data were analyzed with the Vision System software, and only the best value of the 2 eyes was exploited. For each group, 6 control mice and 6 mutant mice were tested.

### 2.5. Retinoid Quantification 

The mice were dark-adapted (DA) for 24 h, and their pupils were dilated with 0.5% tropicamide. The eyes of the mice were enucleated for dark adaptation conditions or exposed to constant light (2 cool white fluorescent lamps, OSRAM of 26 watts, 1800 lm, maximum photopic efficiency of ~470 nm, light intensity averaged 24 mW/cm^2^ corresponding to ≈20.000 lux) during 10 min in a ventilated container, then enucleated for photoadaptation conditions. Throughout the experiment, the ambient temperature was 22 °C, identical to that of the breeding box, to avoid an increase in body temperature, which could further damage the retina. The eyes were rapidly enucleated, frozen in liquid nitrogen, and conserved at −80 °C until use. Retinoids were extracted from eyes following the formaldehyde method with minor modifications [21]. The eyes were homogenized in 200 μL of 3 M formaldehyde/PBS pH 7.5 and incubated for 5 min at 30 °C. Water (100 µL), ethanol (200 µL), and hexane (1.4 mL) were successively added, and the samples were vortexed and then centrifuged at 3000 rpm for 5 min. The upper n-hexane layer was collected, and the extraction was repeated 2 more times. The extracts were combined, evaporated, and dissolved in 20 μL of hexane for HPLC analysis using a Varian HPLC system equipped with a normal phase Nucleodur column (4.6 mm × 250 mm) (Macherey-Nagel) and a Prostar 330 diode array detector. The elution was performed with 6% ethyl acetate in hexane for 10 min at a flow rate of 2 mL/min. The retinoids were analyzed at 365 nm and quantified from the peak areas using calibration curves determined by established standards.

### 2.6. Histochemistry

After sacrifice, the eyes of the mice were enucleated and then immersed in a 4% paraformaldehyde fixative solution (Electron Microscopy Sciences; Hatfield, IN, USA) diluted in PBS (Sigma-Aldrich; Merck; St. Louis, MI, USA) with stirring overnight at 4 °C, then placed in a 70% EtOH (VWR, Rosny-sous-Bois, France, Cat N° 83804.360) solution. Samples were collected and fixed for 24 h in neutral buffered formalin 10%, dehydrated, and embedded in paraffin. Paraffin-embedded tissue was cut into 3 µm thick sections, mounted on slides, and dried at 37 °C overnight. The eyes were washed with cold PBS and depigmented for 30 min with 30% H_2_O_2_ solution (VWR, Rosny-sous-Bois, France, Cat N° 23619.264). Tissue sections were then stained to identify cellular structures, such as nuclei, cytoplasm, and connective tissue with Hematoxylin (Sigma, Saint-Quentin-Fallavier, Cat N° H3136, France), Eosin (Merck, Lyon, France, Cat N° E4382), Saffron (VWR, Rosny-sous-Bois, France, Cat N° 11507737; HES), with HMS 740 autostainer (Microm Microtech, Brignais, France) for preliminary analysis. Collagen fibers were stained with Sirius Red (SR; Merck, Lyon, france, Cat N° 365548), polysaccharides, such as glycogen, glycoproteins, glycolipids, and mucins by Periodic Acid Schiff (PAS; Merck, Lyon, France, Cat N° 3952016), and glycosaminoglycans of the IPM, such as heparan sulfate, proteoglycans, keratan sulfate, hyaluronic acid with Alcian Blue pH 2.6 (AB; Merck, Lyon, France, Cat N° A3157). All the stainings were performed as previously described [22].

### 2.7. Transmission Electron Microscopy (TEM)

At 14 months old, mice were euthanized, and the eyes carefully enucleated. By microdissection, the cornea was removed from the eye. The posterior part of the eye was fixed overnight in 2.5% glutaraldehyde (Electron Microscopy Sciences; Hatfield, IN, USA) and 4% paraformaldehyde (Electron Microscopy Sciences; Hatfield, IN, USA) diluted in PHEM buffer, pH 7.4. Following fixation, the eyes were washed in PHEM buffer, the lens removed, and the eyecups were post-fixed with a solution containing 0.5% osmium tetroxide (Electron Microscopy Sciences; Hatfield, IN, USA) at room temperature for 2 h in the dark. The eyes were rinsed with PHEM buffer and successively underwent ethanol dehydration baths (30–100%; Sigma-Aldrich, Merck; St. Louis, MI, USA). The eyes were then embedded in resin (EmBed 812 Resin; Electron Microscopy Sciences, Hatfield, IN, USA) according to the supplier’s protocol, with the automated microwave for tissue management (EM AMW; Leica; Wetzlar, Germany). After embedding, sagittal sections of 70 nm were cut using an ultra-microtome (Ultracut E; Leica-Reichert; Wetzlar, Germany). The sections were collected and stained with 1.5% uranyl acetate (Electron Microscopy Sciences; Hatfield, IN, USA) in 70% ethanol and lead citrate (Electron Microscopy Sciences; Hatfield, IN, USA). The sections were observed at 120 KV using a transmission electron microscope (Tecnai G2 F20 Spirit BioTWIN TEM; FEI, Thermo Fisher Scientific; Hillsboro, OR, USA) and photographed with a Veleta 4K HS camera. A qualitative comparison was performed between the *Impg1*^−/−^ mice and age-matched control mice of the same genetic background.

### 2.8. Statistics

Statistical analyses were performed using Prism software (GraphPad Software, Inc., La Jolla, CA, USA). The data of two samples were first analyzed with the Shapiro–Wilk normality test, and then two-tailed *p*-values were determined using an unpaired Student’s *t*-test. A *p*-value of <0.05 was considered significant. All analyses were performed on age-matched mutant mice and wild-type littermates. No sex-dependent differences in the phenotypes were noted.

## 3. Results

### 3.1. Generation of an Impg1^−/−^ Murine Model

To investigate the in vivo role of *Impg1* in retinal function, we generated an *Impg1*^−/−^ mouse model by gene targeting. To this end, exon 6 of *Impg1* was floxed in C57BL/6J embryonic stem cells so that the open reading frame would be disrupted in the presence of the Cre recombinase resulting in a premature stop codon (Figure 1A,B). After successive crossings with a CMV-CRE mouse model, we obtained an *Impg1*^−/−^ allele. This allele was then selected by animal crossings to obtain homozygous *Impg1*^−/−^ mice. The *Impg1*^−/−^ mice had no physical or behavioral traits that distinguished them from wild-type mice. To confirm the invalidation of *Impg1*, immunoblotting analysis of whole retina lysates was performed using an anti-SPACR antibody. SPACR protein was detected in the retina of wild-type mice, and a noticeable dose-dependent decrease in its level of production was detected in heterozygous mice, whereas no levels of protein production were detected in *Impg1*^−/−^ mice (Figure 1C). The raw Western blot images are also available in the Appendix A.

### 3.2. Autofluorescent Material Accumulates Subretinally in Impg1^−/−^ Mice

As *IMPG1* was first associated with VMD and, more recently, with RP [11,12,13], we first used in vivo fundus imaging to visualize the retina, the vasculature from the central optic nerve, and the pigmentation of the RPE and choroid (Figure 2A–C). In addition, we used specific filters to detect lipofuscin autofluorescence, which has an excitation spectrum close to 480 nm and an emission spectrum of 575 nm. No defects were observed in the fundus of young mice at 2 months of age (Figure 2A). From 9 months of age, weakly fluorescent white spots were detected throughout the fundus of all *Impg1*^−/−^ mice but not in the wild-type controls (Figure 2B). At 14 months, the spots were intensely fluorescent and covered a large part of the fundus (Figure 2C). These autofluorescent spots in the *Impg1*^−/−^ mice were also detectable by SD-OCT as hyperreflective deposits localized between the photoreceptors OS and the RPE (Figure 2D), but situated closer to the latter. As observed on funduscopy, the multifocal lesions detected by SD-OCT began to appear over the entire retina from approximately 9 months of age, without a preferential localization, and the size and number of lesions increased with age (Figure 2D). Of note, in addition to the small multifocal deposits, a rare, particular lesion was observed in one mouse, a single large dome detected by fundoscopy, SD-OCT, and 3D imaging. This large autofluorescent deposit evokes the yolk-like lesion observable in patients with VMD (Figure 2E).

### 3.3. Subretinal Lesions Lead to a Decrease in Photoreceptor Function

To assess whether the subretinal lesions altered photoreceptor function, we performed ERG recordings. After the animals were dark-adapted, we measured changes in the retinal response to light flash stimulation under scotopic or photopic conditions (Figure 3). No significant change was observed in the scotopic ERG response of the *Impg1*^−/−^ mice versus *Impg1*^+/+^ at 2 months, and a significant difference was seen at 9 months of age (7% decrease in a-wave amplitude for maximum flash intensity; Figure 3A). By contrast, photoreceptor dysfunction was observed at 14 months of age with a 36% decrease in a-wave amplitude and a 31% decrease in b-wave amplitude at maximum light stimulation (Figure 3A,B).

Under photopic conditions, the flicker responses (10 Hz) were significantly reduced by 15% at 9 months and by 43% at 14 months of age in *Impg1*^−/−^ mice compared to control littermates. In addition, we specifically compared the response of S-opsin and M-opsin cones to maximum flash intensity. Decreased responses were only detected in *Impg1*^−/−^ mice from 14 months of age (Figure 3B). As both rod and cone functions were decreased, we evaluated photoreceptor cell death by SD-OCT. We assessed the thickness of the outer retinal layer (ORL), composed of the RPE and photoreceptors cells, compared to the inner retinal layer (IRL), as an internal standard (Figure 3C). This ratio decreased by 18% at 9 months of age and by 26% at 14 months in *Impg1*^−/−^ mice (Figure 3C) as compared to controls.

Next, the isomerization of 11-*cis*-retinal to all-*trans*-retinal after light excitation of the dark-adapted mice was evaluated. The retinoid quantification was performed by high-pressure liquid chromatography (HPLC) of mouse eye extracts. As expected, retinoids extracted from dark-adapted eyes were predominantly 11-*cis*-retinal, and those extracted from light-adapted eyes were predominantly all-*trans*-retinal (Figure 4). No significant difference in retinoid concentration was observed between wild-type and *Impg1*^−/−^ mice at 2 months of age. However, a decrease in the 11-*cis*-retinal concentration of ~40% was observed in 14-month-old dark-adapted *Impg1*^−/−^ mice compared to control mice. A similar decrease in all-*trans*-retinal concentration was observed for the light-adapted 14-month-old *Impg1*^−/−^ mice. Photoisomerization of retinoids appeared to occur normally in *Impg1*^−/−^ mice. Thus, the decrease in basal retinoid concentration in these mice could reflect the loss of photoreceptors from the retina. 

Taken together, the significant decrease in the thickness of the ORL, consistent with the reduced ERG recordings and basal retinoid concentration, suggests photoreceptor degeneration in *Impg1*^−/−^ mice.

### 3.4. Accumulation of Cellular Material in the IPM Leads to Photoreceptor Death

In order to evaluate the retinal structure of the 14-month-old *Impg1*^−/−^ mice, we performed a histological examination and detected a disorganization of the IPM (Figure 5A). The cellular structures, such as the nucleus, cytoplasm, and connective tissue, stained with HES, showed no major retinal defect except at the photoreceptor layer. More specifically, PAS staining of the polysaccharides showed cavities in the photoreceptor OS layer. Sirius Red staining revealed the abnormal presence of collagen around vacuoles inside the IPM (Figure 5B). Alcian Blue staining of the IPM glycosaminoglycans showed that the photoreceptors of the *Impg1*^−/−^ mice were disorganized with the loss of alignment of the photoreceptor OS. Furthermore, the apical microvilli of the RPE appeared less dense than in wild-type mice, and the Bruch’s membrane was abnormal. The microfibrils of Bruch’s membrane were stained but appeared disturbed in *Impg1*^−/−^ mice (Figure 5B). The counting of outer and inner nuclear layer somas was performed on retinal sections of 14-month-old mice stained with Alcian Blue. A decrease in the number of nuclei of 19% is observed in *Impg1*^−/−^ mice as compared to *Impg1*^+/+^ controls (Appendix A).

We also performed ultrastructural studies using transmission electron microscopy (TEM) on the retinas of 14-month-old mice (Figure 6). Compared to the wild-type controls (Figure 6A), defects in the photoreceptor layer were observed in *Impg1*^−/−^ mice (Figure 6B–F). Internal buildup of vacuole-like structures at the photoreceptor-RPE interface was observed (asterisk). Subretinal debris was located between the OS of the photoreceptors and the microvilli of the RPE, mainly in the vicinity of the RPE. Numerous melanosomes were observed among this debris. In addition, many lipid droplets accumulated inside the RPE cells close to where the debris was located (Figure 6C, arrows).

Taken together, the histological and ultrastructure results show that the accumulation of cellular material in the retina of *Impg1*^−/−^ mice results in retinal degeneration and cell death.

## 4. Discussion

### 4.1. Impg1^−/−^ Retinas Accumulate Autofluorescent Material Subretinally

Mutations in the *IMPG1* gene have previously been described in patients with either VMD or RP [11,12]. The underlying molecular mechanisms are very different and particularly complex in these neuropathologies. The lack of pre-existing animal models mimicking a disease phenotype made it difficult to unravel the underlying physiopathological mechanisms associated with *IMPG1* mutations. Thus, in this study, we generated an *Impg1*^−/−^ knockout mouse model to assess the *in vivo* function of the encoded protein SPACR within the IPM and its role in the pathophysiology of IRDs.

Histological and physiological characterizations indicate that homozygous *Impg1*^−/−^ mice develop a late onset and slowly evolving retinal degeneration. Fundus examination revealed numerous white spots scattered all over the retina from 9 months of age but no pigment deposits. These spots represent the accumulation of material deposits and increased in size and number with age. These multifocal lesions were scattered throughout the retina without preference for specific retinal quadrants. These lesions are reminiscent of those observed in VMD patients, although, in humans, they are localized in the macula, a region rich in cones. However, multifocal lesions outside the macula have been observed in patients with *IMPG1*-associated VMD [12]. Furthermore, by in vivo SD-OCT, the lesions in the *Impg1*^−/−^ mice were observed within the IPM, between the photoreceptor OS and the RPE apical microvilli. *IMPG1*-associated VMD patients also present with subretinal material accumulation above the RPE at any stage of the macular dystrophy [13], whereas it should be noted that in the juvenile VMD, Best disease, associated with *BEST1* mutations, the deposits are localized between the RPE and its underlying basement membrane, the Bruch’s membrane [23].

Interestingly, as is the case for VMD [15], the deposits observed in *Impg1*^−/−^ mice were autofluorescent, and the autofluorescence increased strongly with age. Degradation products classically accumulate within the retina with age. The main autofluorescent component of this accumulation is lipofuscin, which, in the human ocular fundus, has a broad band of emission between 500 and 750 nm, a maximum of approximately 630 nm, and optimal excitation of approximately 510 nm [24]. This is consistent with the excitation and emission wavelengths observed in the *Impg1*^−/−^ mice.

### 4.2. Subretinal Material Leads to Moderate Physiological Impairment

With age, the *Impg1*^−/−^ mice showed a reduction in the ERG response. An electrophysiological study showed that the damage was moderate and that the rods and cones were affected approximately equally. Likewise, the two cone subtypes, S- and M-opsin, showed similarly reduced responses. The decrease in rod and cone ERG responses was correlated with photoreceptor degeneration, as determined by in vivo SD-OCT showing a 26% decrease in the thickness of the ORL layer at 14 months and by histological staining showing a 19% decrease in the number of photoreceptor nuclei at 14 months. Likewise, a decrease in 11-*cis*-retinal concentration of about 40% observed at 14 months is also in favor of this loss of photoreceptors. We hypothesize that the dysfunctional photoreceptors, regardless of the type, in *Impg1*^−/−^ mice are located adjacent to the accumulated material and disorganized IPM.

Previous in vivo studies on the *Impg2*^−/−^ mouse model showed weakened ERG responses, with retinal lesions similar to those detected in our *Impg1*^−/−^ model [17,18]. Surprisingly, Salido et al. generated *Impg1*^−/−^, *Impg2*^−/−,^ and double mutant mice and found no evidence of morphological or functional deficits in the *Impg1*^−/−^ model and only small deficits in the double mutant mice [18]. We can make the following speculations about this apparent contradiction with our study. Firstly, the *Impg1*^−/−^ animals in the study by Salido et al. were younger than 8 months when they were phenotyped by ERG. Secondly, the authors did not perform *in vivo* fundus imaging to visualize the retina. Therefore, they did not detect the white spots, which constitute the first signs of degeneration in the absence of *Impg1*. The retinal degeneration of the *Impg1*^−/−^ mouse is not detectable until 9 months of age either by ERG or funduscopy. This late-onset and moderate phenotype in the mutant mice is consistent with the clinical findings in human patients who have an adult-onset *IMPG1*-associated VMD form [11,12]. Likewise, for RP, *IMPG1* variants are associated with an overall moderate form of the disease with asymptomatic carriers [11].

### 4.3. IPM Disorganization Leads to Alteration of the RPE

Histological studies of the *Impg1*^−/−^ retina showed a disorganized arrangement of photoreceptor cells with, in particular, a shape alteration by the loss of alignment of the photoreceptor OS. In addition to misaligned photoreceptors, vacuoles are also visible at the IPM in all *Impg1*^−/−^ animals. Specific staining of the collagen fibers by the Sirius Red dye was detected in a few vacuoles, in addition to the physiological staining of RPE and Bruch’s membrane. We, therefore, hypothesized that the integrity of the apical RPE membrane was disrupted in *Impg1*^−/−^ mice, and the cytoplasmic content was released into the IPM. Alcian Blue was used to stain mucosubstances in the IPM (glycoproteins and proteoglycans) [25]. Surprisingly, the entire OS layer is disorganized in the mutant mice, and not just a few limited areas. Likewise, RPE microvilli and Bruch’s membrane staining also appeared to be weaker, revealing a loss of integrity of these structures in the absence of SPACR. Furthermore, TEM analysis identified much debris, including melanosomes, at the interface of RPE and photoreceptors OS, thus highlighting morphological defects of the RPE. In addition, lipid droplets accumulated inside the RPE cells indicating a visual cycle defect in the RPE [26].

We, thus, hypothesize that disorganization of IPM due to lack of SPACR leads to dysfunction and alteration of the RPE. Phagocytosis occurs by a first step of recognition and attachment of the OS with the RPE through the IPM [27]. A defect of the IPM could alter this first step and then disrupt the phagocytosis of shed photoreceptor OS, overloading the RPE cells over time with degraded material. Lipofuscin in RPE represents an altered form of the digested material and plays an important role in the progression of VMD. Indeed, VMD is associated with abnormal accumulation of lipofuscin both above, within, and below the RPE. Lipofuscin cytotoxicity would then result in photoreceptor degeneration.

In summary, *Impg1*^−/−^ mice exhibit late and moderate alteration of the retina and photoreceptor activity, which was not observed in a previous study due to the young age of the animals studied. Subretinal autofluorescent lesions appear from 9 months of age, accompanied by a disorganized arrangement of the photoreceptors and the alteration of the RPE. Thus, the *Impg1*^−/−^ mouse is an informative model for deciphering the mechanisms underlying *IMPG1*-associated IRDs and for developing novel therapies.

## Figures and Tables

**Figure 1 genes-13-01508-f001:**
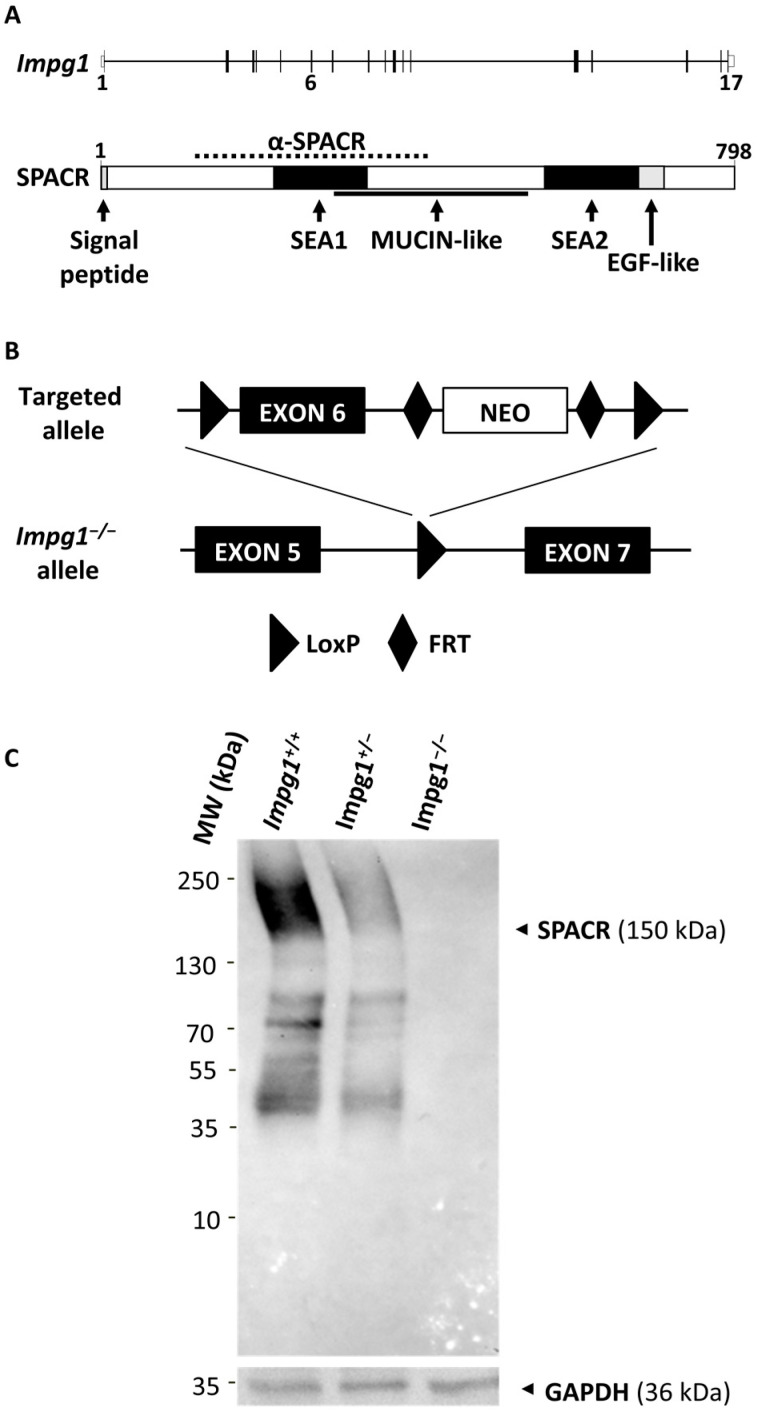
Validation of the *Impg1*^−/−^ mouse model. (**A**) Schematic representation of the exon-intron (exons 1 to 17) structure of *Impg1* and the domain structure of its encoded SPACR protein (amino acids 1 to 798), including the signal peptide, the N-terminal SEA1 and C-terminal SEA2 domains, the mucin-like domain, and the EGF-like domain. The anti-SPACR mouse monoclonal antibody (α-SPACR) was raised against amino acids 121–420 (dotted line). (**B**) The exon 6 (black box) of *Impg1* mouse was floxed in C57BL/6J embryonic stem cells to disrupt the open reading frame. Arrowheads indicate the loxP sites of the Cre/loxP system. The positive selection marker neo is flanked by FRT sites (diamonds). (**C**) The expression of SPACR protein in wild-type (*Impg1*^+/+^), heterozygote (*Impg1*^+/−^) and SPACR-deficient (*Impg1*^−/−^) whole retina lysates was analyzed by Western blotting using a mouse monoclonal antibody against SPACR. GAPDH expression was used as protein loading control. MW, molecular weight in kilodalton (kDa).

**Figure 2 genes-13-01508-f002:**
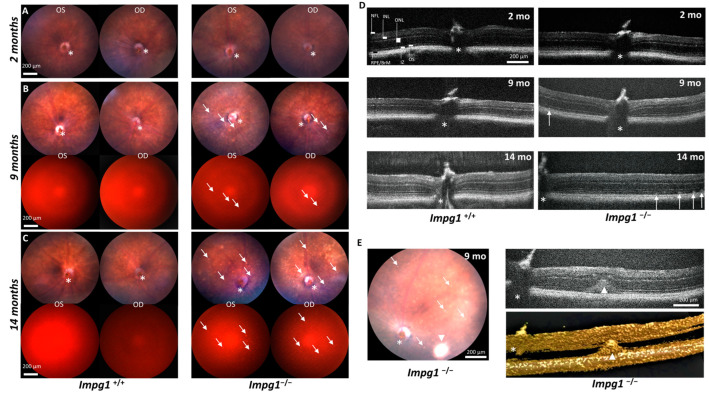
*Impg1*^−/−^ mice develop autofluorescent subretinal deposits. (**A**) No defects were observed in the fundus of young mice of 2 months of age. Asterisks indicate the optic nerve (*n* = 10). (**B**) Fundus examination showed numerous retinal hypopigmented deposits (arrows) from 9 months of age, some of which are autofluorescent (excitation range is 444.5–564.5 nm and emission range > 568 nm). Asterisks indicate the optic nerve (*n* = 10). (**C**) Fundus examination of 14-month-old mice. Retinal deposits (arrows) increase in number and autofluorescence with age (*n* = 10). (**D**) SD-OCT localized these hyper-reflective clusters between the photoreceptor OS and the RPE. Ten wild-type mice and 10 *Impg1*^−/−^ mice were examined, and all 9-month and 14-month-old mutant mice had similar clusters (white arrows) scattered all over the retina, which were not present in wild-type mice (top, left-hand panel). (**E**) Funduscopy and SD-OCT imaging of an *Impg1*^−/−^ retina with a large localized autofluorescence nummular clumping (arrowheads). Lower right panel shows 3D reconstitution. NFL, nerve fiber layer; INL, inner nuclear layer; ONL, outer nuclear layer; RPE, retinal pigment epithelium; BrM, Bruch’s membrane; IZ, interdigitation zone; OS, outer segment.

**Figure 3 genes-13-01508-f003:**
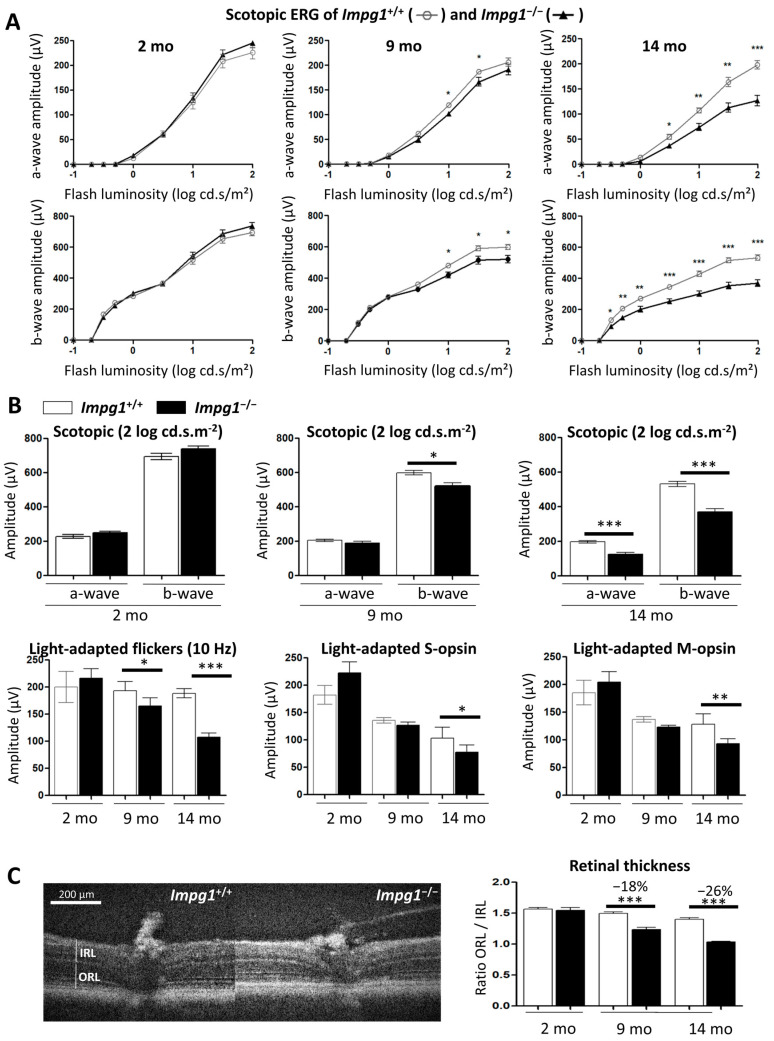
Late functional deficits in rods and cones in *Impg1*^−/−^ mice. (**A**) Scotopic ERG response of *Impg1*^−/−^ mice (black triangles, thick line) compared to wild-type mice (white circles, thin line) as a function of flash intensity. Bars represent mean ± SEM, *n* = 6 for each time point (2, 9, and 14 months) and for each condition, with normal distribution, Student’s *t*-test was used. * *p* < 0.05, ** *p* < 0.01, *** *p* < 0.001. (**B**) *IMPG1* knockout (black bar) resulted in reduced amplitudes of the scotopic (upper left panel) and photopic responses at light intensity levels of 2 log cd.s.m^−2^ and 10 Hz flicker ERG (upper right panel). Opsin S (blue light) and M (green light) responses are decreased in *Impg1*^−/−^ mice (lower panels). Bars represent mean ± SEM, *n* = 6 with normal distribution; Student’s *t*-test was used. * *p* < 0.05, ** *p* < 0.01, *** *p* < 0.001. (**C**) SD-OCT analysis of *Impg1*^+/+^ versus *Impg1*^−/−^. Left panel shows the retinal section of a wild-type mouse with annotated layers. The thickness of the inner retinal layer (IRL) and outer retina layer (ORL) were measured for each genotype, and the ORL/IRL ratio was used to evaluate outer retina thickness in wild-type (*Impg1*^+/+^, white bars) vs. mutant (*Impg1*^−/−^, black bars) mice at 2, 9, and 14 months of age (right panels). The ORL/IRL ratio was reduced in *Impg1*^−/−^ mice. Bars represent the mean of 10 OCT measurements per animal, 5 animals per group; Student’s *t*-test; *** *p* < 0.001.

**Figure 4 genes-13-01508-f004:**
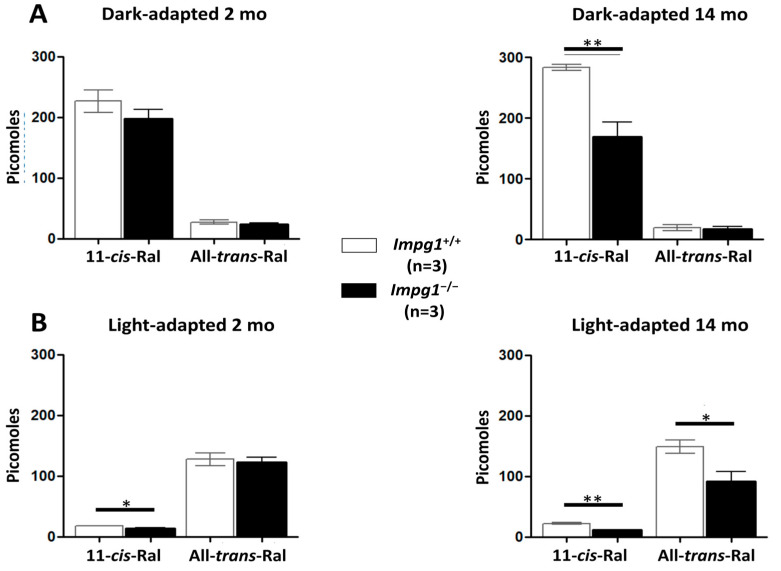
Decrease in retinoid concentration in *Impg1*^−/−^ mice. Under dark-adapted conditions, the 11-*cis*-retinal (11-*cis*-Ral) concentration is maximal (**A**), whereas the all-*trans*-retinal (All-*trans*-Ral) concentration is maximal after 10 min of constant light exposure (**B**). *Impg1*^−/−^ mice seem to show normal photoisomerization, but the basal retinoid concentration is 40% lower than wild-type mice at 14 months of age under light exposure. Student’s *t*-test (*n* = 6 eyes); * *p* < 0.05, ** *p* < 0.01.

**Figure 5 genes-13-01508-f005:**
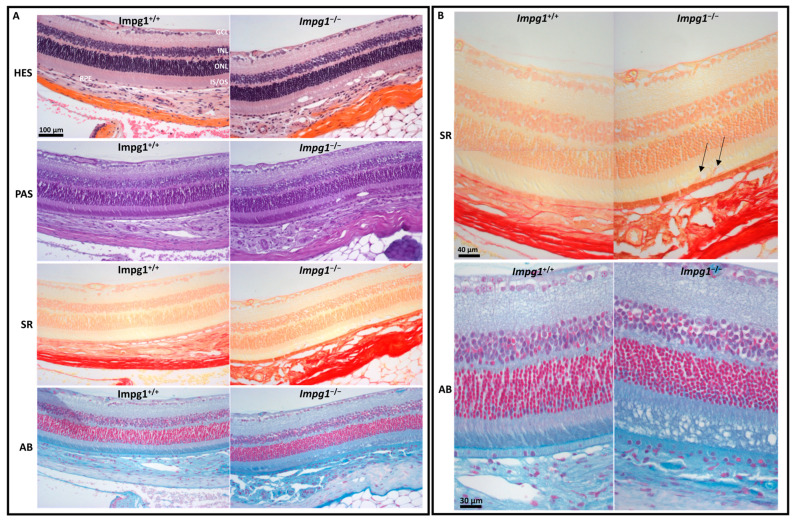
Disorganization of the IPM. (**A**) Histological staining of 14-month-old mouse retina showing IPM disorganization in the *Impg1*^−/−^ mice. Retinal sections were stained with Hematoxylin/Eosin/Saffron (HES), Periodic Acid Schiff (PAS), Sirius Red (SR), and Alcian Blue (AB) to evaluate the nature of the material abnormally localized in the IPM; magnification ×20. (**B**) Sirius Red confirms the presence of collagen around the vacuoles (arrows) in the IPM of *Impg1*^−/−^ mice compared to wild-type retina. Alcian Blue staining shows the loss of the vertically straight orientation of the *Impg1*^−/−^ photoreceptors, the presence of numerous vacuoles mainly in the outer segments layer.

**Figure 6 genes-13-01508-f006:**
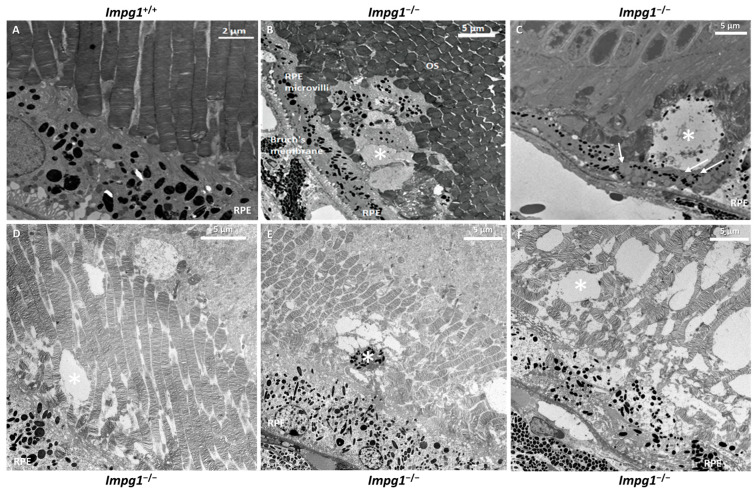
Abnormal accumulation of cellular material in the IPM. Transmission electron microscopy (TEM) on the retinas of 14-month-old wild-type *Impg1*^+/+^ control (**A**) and *Impg1*^−/−^ (**B**–**F**) mice. In the absence of SPACR protein, internal buildup of vacuole-like structures at the photoreceptor-RPE interface was observed (asterisk). In addition, an accumulation of lipid droplets can be seen in the RPE cells (arrows) in contact with this material.

## Data Availability

The data presented in this study are available on request from the corresponding author.

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
