# Peer review of "SPACR Encoded by IMPG1 Is Essential for Photoreceptor Survival by Interplaying between the Interphotoreceptor Matrix and the Retinal Pigment Epithelium"

_genes, 2022, doi:10.3390/genes13091508_

Round 1

Reviewer 1 Report

Major comments:

The authors describe the retina-related changes of an Impg1 knockout. Although the development of new animal models depicting human diseases is important to develop new treatments, presentation of the data in the manuscript is mediocre. Especially, the depicted time points of Figures 3 and 4 are confusing. Why did the authors not show always month 2, 9, and 14 as described in M&M and Figure 3A? In addition, the light-adapted ERG results in Figure 3B have to be extended corresponding to 3A. Furthermore, magnification of Figure 5A is far too low in order to acknowledge anything that is described in the text. Thickening of the Bruch’s membrane has to be quantified in order to be mentioned (Figure 5 legend). In the discussion, the authors do a lot of speculations, which is not increasing the plausibility of the manuscript.

Minor comments:

P5 line 141: As the changes at 9 months are already significant, I would not name them slight.

P5 line 159: In Figure 4B the titles should be “light-adapted” as in the corresponding text.

Figure 2: Asterisks are missing in panel C. Explanation of the arrowhead in E is missing in the legend. Add “lower” before “Right panel” in E in the legend.

Figure 6: What is the difference between the scale bars in the panels and the ones mentioned in the legend?

Reviewer 2 Report

In this manuscript entitled the Authors study the effect of knocking out IMPG1, that encodes interphotoreceptor matrix proteins SPACR on the morphology and physiological parameters of mouse retina, that are known to be important in photoreceptor maturation. Importantly, mutations of IMPG1 have been associated with vitelliform macular dystrophy in humans, however, a recently generated Impg1-/- mouse model (or the Impg1-/-; Impg2-/- double knockout) did not show pathological signs. Thus, the objective of the current investigation was to generate a new Impg1-/- mouse line and study if pathological phenotype is observed.

The purpose is legit, the research is important, and the utilized methodology to study the transgenic line makes perfect sense. The results brought about by the various experimental methods support each other by demonstrating that the Impg1-/- mice gradually develop a structural disorganization in their retina, primarily affecting the outer segments of photoreceptors. The reported structural disorganization is associated by visual pigment deficits as well as visual function decay as evidenced by ERG measurements. This is a data rich, carefully written manuscript. I think the conclusion (i.e. “the Impg1-/- mouse is an informative model for deciphering the mechanisms underlying IMPG1-associated IRDs and for developing novel therapies”) is supported by the presented data.

I have two comments for improvement:

1.       While the SD-OCT analysis of retina thickness shows significant outer retina thinning/degeneration of the photoreceptor layer in the Impg1-/- mice, photoreceptor loss per say has not been explicitly investigated, even though extensive histological analysis was performed. It would be very informative, if at least ONL soma counts would be done on representative HES stained retinal sections (i.e. over 100 µm length, Impg1-/- vs. control) to see if, and how extensive complete photoreceptor loss is taking place in the Impg1-/- mice by the 14th  month; alternatively, the strong visual deficits detected by ERG at this age are simply caused by the misalignment and decay of outer segments and/or by disrupted RPE/outer segment interactions. Considering the human pathology seen in VMD and RP, this information could elucidate an important detail of the Impg1-/- mice as a model.

2.       Please communicate the appropriate statistics used for ERG data comparison between control vs. Impg1-/- (Figure 3A) either in the Methods or in the corresponding figure legend.

Round 2

Reviewer 1 Report

All comments are answered in the new version.